

# Transcriptome-wide characterization and functional analysis of MATE transporters in response to aluminum toxicity in *Medicago sativa* L.

Xueyang Min[1,2,3,4], Xiaoyu Jin[1,2,3,4], Wenxian Liu[1,2,3,4], Xingyi Wei[1,2,3,4], Zhengshe Zhang[1,2,3,4], Boniface Ndayambaza[1,2,3,4] and Yanrong Wang[1,2,3,4]

[1] State Key Laboratory of Grassland Agro-Ecosystems, Lanzhou, P. R. China
[2] Key Laboratory of Grassland Livestock Industry Innovation, Ministry of Agriculture and Rural Affairs, Lanzhou, P. R. China
[3] Engineering Research Center of Grassland Industry, Ministry of Education, Lanzhou, P. R. China
[4] College of Pastoral Agriculture Science and Technology, Lanzhou University, Lanzhou, P. R. China

Corresponding authors
Wenxian Liu, liuwx@lzu.edu.cn
Yanrong Wang, yrwang@lzu.edu.cn

## ABSTRACT

Multidrug and toxic compound extrusion (MATE) transporters contribute to multidrug resistance and play major determinants of aluminum (Al) tolerance in plants. Alfalfa (*Medicago sativa* L.) is the most extensively cultivated forage crop in the world, yet most alfalfa cultivars are not Al tolerant. The basic knowledge of the MATE transcripts family and the characterisation of specific MATE members involved in alfalfa Al stress remain unclear. In this study, 88 alfalfa MATE (MsMATE) transporters were identified at the whole transcriptome level. Phylogenetic analysis classified them into four subfamilies comprising 11 subgroups. Generally, five kinds of motifs were found in group G1, and most were located at the N-terminus, which might confer these genes with Al detoxification functions. Furthermore, 10 putative Al detoxification-related *MsMATE* genes were identified and the expression of five genes was significantly increased after Al treatment, indicating that these genes might play important roles in conferring Al tolerance to alfalfa. Considering the limited functional understanding of MATE transcripts in alfalfa, our findings will be valuable for the functional investigation and application of this family in alfalfa.

## INTRODUCTION

Aluminum (Al) is the third most predominant inorganic monomeric component in the outside layer. Roughly 30% of the ice-free land area comprises the best soil with a pH < 5.5, and half of the world's conceivably arable grounds are acidic (*Kochian, Hoekenga & Piñeros, 2004*; *Vonuexkull & Mutert, 1995*). Unlike other metals, Al toxicity is much more threat to plant grown in acidic soil, and it remains quite toxic to plant roots and

inhibits root elongation, further leading to roots stunting accompanied by reduced water and nutrients uptake (*Kochian, Hoekenga & Piñeros, 2004*; *Wood et al., 2000*). For Al toxicity, two mechanisms that encourage Al tolerance in plants are preventing Al particles from entering the roots (exclusion mechanism) and detoxifying internal Al in the symplast (tolerance mechanism) (*Ma, Ryan & Delhaize, 2001*; *Sade et al., 2016*; *Zheng et al., 2005*). Many Al-accumulating plants, such as *Melastoma malabathricum* and buckwheat, possess the inner Al detoxification ability to form the development of metal and accumulating them in the above-ground herbage (*Ma, Ryan & Delhaize, 2001*; *Zheng et al., 2005*). Another mechanism of Al tolerance in plants involves Al initiation of membrane transporters that intercede organic acid (OA; such as malate, citrate, and oxalate) exclusion from the root apex and form non-toxic complexes with rhizosphere aluminium (*Kochian, Hoekenga & Piñeros, 2004*; *Sade et al., 2016*). The mechanism of Al tolerance has reported in many crops, such as rice, wheat, soybean, maize, sorghum, buckwheat, and rye (*Kochian, 1995*; *Kochian et al., 2005*; *Ma, Ryan & Delhaize, 2001*; *Zheng et al., 2005*). Recently, the root apex has also recommended to role in Al tolerance, which is related to serious changes in the root system, including cell differentiation in root tips and lateral roots, increasing cell wall inflexibility, interfering with several enzymes, reducing DNA replication, modifying the structure and capacity of plasma membranes, and disrupting signal transduction pathways (*Sade et al., 2016*; *Zheng et al., 2005*).

The citrate transporter is an electrochemical potential-driven transporter activated by Al and belongs to the multidrug and toxic compound extrusion (MATE) family (*Yokosho et al., 2016*). A *MATE* gene (*Norm*) was cloned in *Vibrio parahaemolyticus*, which possesses an energy-dependent system that effluxes nofloxacin and other antimicrobial agents outside of the cells by $Na^+$/drug antiport (*Morita et al., 1998*). The MATE family consists of a unique topology predicted to have 10–12 transmembrane (TM) helices with long, cytoplasmatic C and N termini (*Darban et al., 2016*). To date, 56, 53, 117, 70, 49, 70, and 68 MATE transporters have been identified from *Arabidopsis* (*Li et al., 2002*, *2009*), *Oryza sativa* (*Tiwari et al., 2014*), *Glycine max* (*Liu et al., 2016*), *Medicago truncatula* (*Wang et al., 2017*), *Zea mays* (*Zhu et al., 2016*), *Gossypium raimondii* and *Gossypium arboreum* (*Lu et al., 2018*), respectively. According to the established MATE functions, three major classes associated with disease resistance, small organic molecule exportation, and secondary metabolite transportation have been classified (*Diener, Gaxiola & Fink, 2001*; *Hiasa et al., 2006*; *Tiwari et al., 2014*). As the first MATE transporter identified in plant, AtALF5 (*Arabidopsis* aberrant lateral root formation 5) was involved to the protection of roots from multidrug resistance (*Diener, Gaxiola & Fink, 2001*). AtDTX1 (*A. thaliana* detoxification 1) was characterised as an efflux transporter for plant-inferred alkaloids, anti-infections, and different exogenous toxic compounds (*Li et al., 2002*). MATEs have also been demonstrated to be associated with plant disease resistance, such as enhanced disease susceptibility 5 (EDS5), which functions in reducing basal resistance during pathogen interaction (*Nawrath et al., 2002*). OsMATE1 and OsMATE2 negatively alter stress responses and pathogen vulnerability (*Tiwari et al., 2014*). Additionally, different research studies have also showed that MATE transporters

mediate citrate efflux to present plant resilience and tolerance to Al toxicity (*Magalhaes et al., 2007*; *Wood et al., 2000*). Moreover, roles of some genes in this family in tolerance towards Al toxicity in acidic soils, which involve an electrochemical slope of cations (such as $H^+$ or $Na^+$ ions) over the film to drive substrate export, have been examined (*Shoji, 2014*). The first aluminum-activated citrate transporter, SbMATE (*Sorghum bicolor* MATE), was characterised in sorghum (*Magalhaes et al., 2007*). Similar to this protein from sorghum, MATE proteins from *Hordeum vulgare* (HvAACT1) (*Furukawa et al., 2007*), *O. sativa* (OsFRDL4) (*Yokosho, Yamaji & Ma, 2011*), *Zea mays* (ZmMATE1) (*Maron et al., 2010*) and *Eucalyptus camaldulensis* (EcMATE1) (*Sawaki et al., 2013*) have additionally been appeared to take an interest in Al resilience by transporting organic acid anions from roots to decrease Al absorption in the rhizosphere (*Ma, 2000*; *Yokosho et al., 2016*).

As the most widely grown perennial legume crop, alfalfa provides numerous agro-ecological advantages, including a high nutritional value, a perennial high yield, contributions to soil fertility, and multiple harvests during the growing season (*Min et al., 2017*; *Wang et al., 2016a*). However, it is responsive to soil acidity, and its production and stand duration are severely impaired by acidic soils due to root growth inhibition and weakened nitrogen fixation (*Hartel & Bouton, 1991*; *Liu et al., 2017*). While the existing research on MATE transporters is mostly focused on model plants, the information regarding the expression and function of *MATE* genes related to Al stress in alfalfa are still not yet reported. Here, we systematically identified the MATE family at the transcriptome-wide level using a rigorous method to provide basic data regarding alfalfa MATE transporters. Our observations provided a first sight of how these Al-tolerant regulators behave when alfalfa is disputed by Al stress. This fundamental research would provide the valuable information for the further gene cloning and biotechnology studies of the MATE transporters in alfalfa.

# MATERIALS AND METHODS

## Plant growth and Al treatment

The seeds of alfalfa (cultivar Zhongmu No.1) were surface sterilized in 1.0% (v/v) sodium hypochlorite for approximately 5 min and then tap water washed three times and germinated at 20 °C in the dark. After 3 days, uniform seedlings were transplanted to half-strength modified Murashige and Skoog solution (pH 5.6) in plastic containers, and the solution was changed every 2 days. The seedlings were grown in a growth chamber under controlled conditions (light intensity 120 $\mu$mol m$^{-2}$s$^{-1}$, 16/8 h light/dark cycle at 25 °C and a relative humidity of 65%). After cultivating for 7 days at 25 °C with a photoperiod of 16 h light/8 h dark, the seedlings were transferred to 0.5 mM CaCl$_2$ (pH 4.5) for 24 h before Al treatment. The seedlings were then exposed to 0.5 mM CaCl$_2$ (pH 4.5) solution containing either zero $\mu$M AlCl$_3$ (control) or 20 $\mu$M AlCl$_3$ (treatment) for 4, 8, and 24 h, respectively. The root tips (approximately two cm in length) were collected, both the main root tips and the secondary root tips, and immediately frozen in liquid nitrogen and stored at −80°C. The experiment was performed in triplicate.

## Sequences dataset and MATE transporter identification in alfalfa

To identify novel alfalfa MATE transporters, all the alfalfa annotated unigenes were downloaded from the Alfalfa Gene Index and Expression Atlas Database (AGED) (*O'Rourke et al., 2015*). *Arabidopsis* (57), rice (52), *M. truncatula* (70), and soybean (117) MATE protein sequences were collected from Phytozome v12 (*Goodstein et al., 2012*). Basic Local Alignment Search Tool algorithms were applied with the published *O. sativa*, *Arabidopsis*, *M. truncatula* and soybean MATE transporters as a query to search against the AGED database and with *e*-value cutoff set as 1e−5. The conserved protein domain of these candidates was checked using NCBI-CDD search (http://www.ncbi.nlm.nih.gov/Structure/cdd/wrpsb.cgi). If an open reading frame (ORF) contained the complete MATE domain, then coding sequence (CDS) were manually retrieved with ORF Finder (http://www.ncbi.nlm.nih.gov/gorf/gorf.html) and translated into protein sequences in Sequence Manipulation Suite (http://www.bioinformatics.org/sms2/translate.html). All the obtained protein sequences were examined for the presence of MATE domains using the Pfam (http://pfam.sanger.ac.uk/search) tool. Multiple alignments were also performed to avoid repetition.

## Phylogenetic tree construction and conserved motifs identification

A bootstrap neighbour-joining (NJ) phylogenetic tree was constructed among alfalfa, *Arabidopsis,* and rice MATEs using ClustalW in Clustal Omega (*Mohanta et al., 2015*; *Sievers et al., 2011*), then the phylogenetic tree data was used to construct a NJ tree in MEGA 7.0 (*Kumar, Stecher & Tamura, 2016*). Motifs of MsMATE proteins were determined using the Multiple Expectation Maximization for Motif Elicitation (MEME) (http://meme-suite.org/), and a schematic diagram of amino acid motifs of each MsMATE protein was drawn accordingly with the default parameters, except the maximum number of motifs to find was set to 12 (*Bailey et al., 2015*).

## In silico sequence analysis

The theoretical isoelectric point (pI) and molecular weight (Mw) of the MsMATE proteins were calculated using the Param tool (http://web.expasy.org/protparam/), and subcellular localisation was predicted using WoLF PSORT (*Horton et al., 2007*). Number of TM helices was predicted using TMHMM Server v. 2.0 (*Krogh et al., 2001*). *Arabidopsis* orthologues for alfalfa MATE proteins were identified by a BLASTP search against *Arabidopsis* proteins TAIR10 release (http://www.arabidopsis.org). The functional interacting networks of predicated Al detoxification MATE proteins were integrated using STRING software with the confidence limit set at 0.400 (*Von Mering et al., 2005*). The expression data was retrieved from the AGED database (NCBI accession: SRP055547), and the expression data were gene-wise normalized using the MeV v4.9 software (http://www.mybiosoftware.com/).

## RNA extraction and qRT-PCR analysis

Total RNA was extracted from tissues using the Sangon UNIQ-10 column TRIzol total RNA extraction kit and digested using DNase I to eliminate genomic DNA contamination

according to the manufacturer's instructions. Subsequently, first-strand cDNA was synthesized from four μg of total RNA by M-MuLV reverse transcriptase (Sangon Biological Engineering Technology & Services, Shanghai, China). Gene-specific primers (Table S1) for qRT-PCR analysis were designed using NCBI Primer-BLAST and their specificity was checked by blasting each primer sequence against the alfalfa transcriptome sequence. The qRT-PCR analysis was performed in three technical replicates using the 7500 Fast Real-Time PCR system (Applied Biosystems, Foster City, CA, USA). The 20 μL qRT-PCR mixtures included 10 μL of 2 × SG Fast qPCR Master Mix (Low Rox), 0.5 μL of forward and reverse primers, two μL of DNF Buffer, one μL of diluted cDNA solution, and six μL of ddH$_2$O. The thermal profile for qRT-PCR was as follows: 10 min at 95 °C for DNA polymerase activation, followed by 40 cycles of 15 s at 95 °C and 1 min at 60°C. As an internal standard, *Ms-Actin* was used to calculate the relative fold differences based on the comparative *Ct* method. The expression levels were calculated using the $2^{-\Delta\Delta CT}$ method.

## RESULTS

### Identification of MsMATE transporters

To identify the *MsMATE* transporters in alfalfa, the previously identified MATE proteins in several model plant species were used as a query to search the alfalfa transcriptome dataset. A total of 106 contigs with complete conserved domains and full-length cDNA sequences were identified (Table S2). After removing redundancy, 88 unique *MsMATE* genes finally remained for further phylogenetic and functional analysis (Table S3; Supplementary Data 1). According to their numeric sorting, the 88 genes were designated *MsMATE01–MsMATE88*. The ORFs of these genes were varied in length from 321 (MsMATE14) to 1,788 (MsMATE20) bp, and the predicted protein products ranged from 106 to 595 amino acids of length with Mw varying from 11.61 to 65.3 kDa, containing 0–12 TMs, the protein grand average of hydropathy varying from 0.358 (MsMATE06) to 1.159 (MsMATE22). The protein isoelectric point varied from 4.51 (MsMATE12) to 10.33 (MsMATE03), suggesting that these MATE proteins may work under different conditions. In addition, 63 plas (plasma membrane, 71.60%), 10 chlo (chloroplast, 11.36%), six cyto (cytoplasm, 6.82%), seven vacu (vacuole, 7.95%) and two nucl (nucleus, 2.27%) were found according to the subcellular location results (Table S3).

### Phylogenetic relationship analysis of MsMATE transporters

To categorise and investigate the evolutionary relationships among different species, the MATE proteins in alfalfa, *Arabidopsis,* and rice were chosen to perform phylogenetic scrutiny (Fig. 1; Supplementary Data 2). Based on DNA-binding motifs and functional properties, all these MsMATE domains could be classified into four main classes including 11 smaller subgroups (Fig. 1), namely, class I (subgroups CI-1, CI-2, and CI-3), class II (subgroups CII-1 and CII-2), class III (subgroups CIII-1, CIII-2, and CIII-3) and class IV (subgroups CIV-1, CIV-2, and CIV-3).

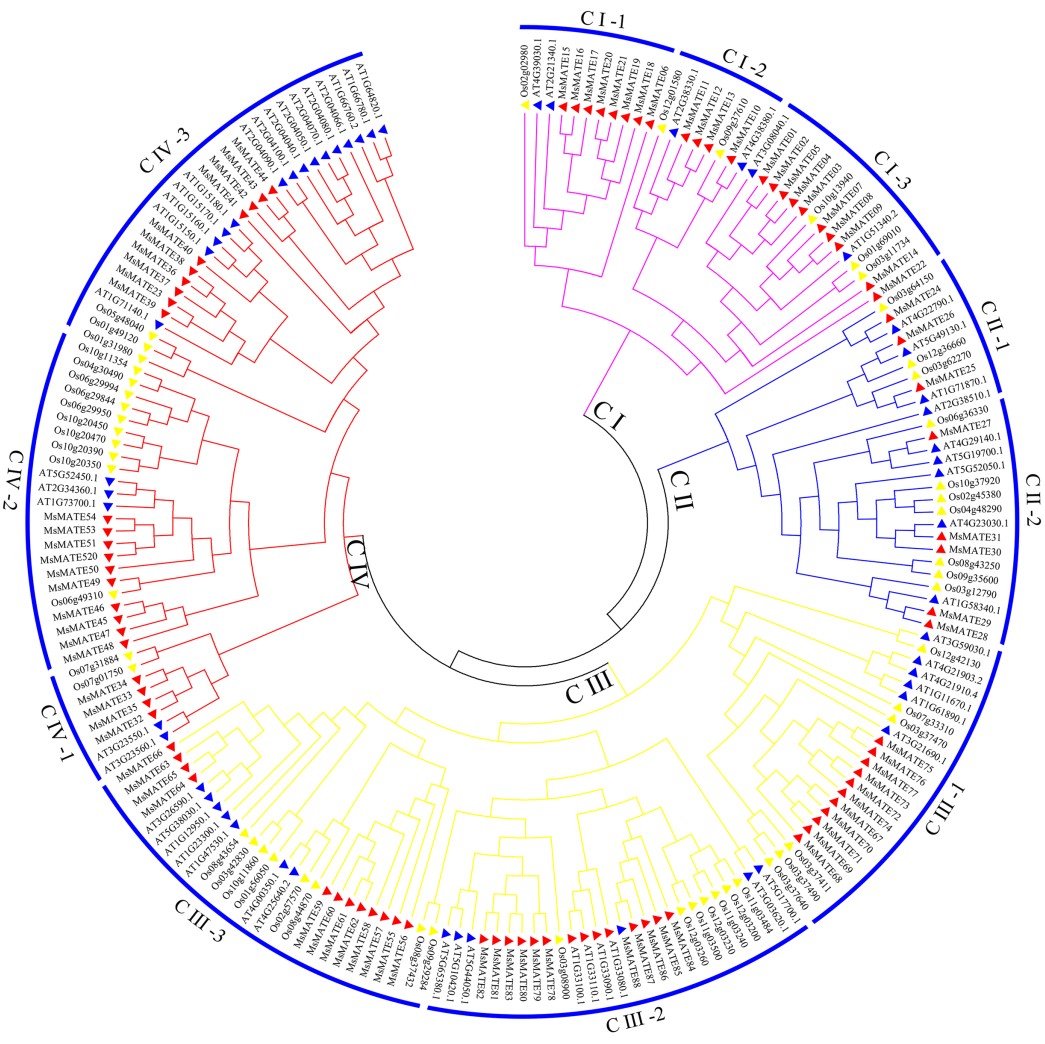

**Figure 1 Phylogenetic tree of MATE proteins from *Arabidopsis*, rice, and alfalfa.** The phylogenetic tree was constructed using MEGA7.0 and the neighbour-joining method with 1,000 bootstrap replicates. The tree was divided into four subfamilies comprising 11 smaller subgroups. Members of *Arabidopsis*, alfalfa, and rice are denoted by red, blue, and yellow triangles, respectively.

To similarly predict and distinguish the feature of identified *MsMATE* genes in this study, an NJ tree was constructed using 88 MsMATE proteins and 38 MATE transporters from other species, which have been well investigated at the molecular level and proved have specific function (Fig. 2; Table S5). Four main distinct groups (groups G1, G2, G3, and G4) comprising 12 smaller subgroups were finally identified (Fig. 2). The first group consist of 22 MsMATEs, and some MATE transporters function as ligand and molecules to bind Al detoxification/iron translocation in the rhizosphere. In addition, four subgroups within the first group (G1) were observed. A total of 10 MsMATE proteins and all the other 19 Al detoxification/iron translocation-related genes were classed into subgroups G1–3, indicating that these 10 MsMATE proteins might be concerned in Al detoxification and/or iron translocation in alfalfa. Previous lookup located that

AtADS1/ABS3 is a negative regulator of plant disease resistance (*Sun et al., 2011*) and that ELS1 is a novel MATE transporter associated to leaf senescence and iron homeostasis in *Arabidopsis* (*Wang et al., 2016b*). Overexpression of AtZF14/BCD1/ABS4 and AtADS1/ABS3 in *Arabidopsis* suggested that these *MATE* genes function at an increasing organ initiation rate, sustaining iron homoeostasis and hypocotyl cell elongation (*Burko et al., 2011*; *Seo et al., 2012*; *Wang et al., 2015*). In our research, there are five MsMATE proteins classified into G2-2 with the other three MATE proteins (AtADS1, ELS1, and AtZF14), G2 subgroup may has diverse physiological functions in alfalfa. Group G3 (24 MsMATE proteins) could be further classified into three subgroups. Subgroup G3-1 has four MsMATE proteins and a recognized MATE protein AtALF5 (*A. thaliana* aberrant lateral root formation 5), which is expressed strongly in the root epidermis and expanded the sensitivity of roots to a variety of compounds (*Diener, Gaxiola & Fink, 2001*). Subgroup G3-2 incorporates nine MsMATE members, with no formerly acknowledged MATE proteins. There are 10 MsMATE proteins in the G3-3 subgroup and two in the past mentioned MATE transporters, namely, NtJAT1 (*N. tabacum* jasmonate-inducible alkaloid transporter 1) and AtDTX1. Previous studies have showed that NtJAT1 used to be determined to characteristic as a secondary transporter for nicotine translocation in tobacco (*Morita et al., 2009*), and AtDTX1 mediated plant-derived efflux and detoxified the heavy metal ($Cd^{2+}$). It also served to efflux other toxic compounds (*Li et al., 2002*). Therefore, subfamily G3 might be related to the xenobiotics and alkaloid transporter. Among all the groups, G4 was once the biggest clade (including 34 MsMATEs) and ought to be further divided into three subgroups. The functions of the recognised MATE transporters in this group are concerned in trafficking of secondary metabolites in vacuoles and accumulating flavonoids or anthocyanin in plants. For example, the *Arabidopsis* TT12 transporter exhibits extended substrate specificity accepting glycosylated anthocyanidin in vitro (*Marinova et al., 2007*), and *AtFFT* is highly transcribed in floral tissues and affects flavonoid levels in *Arabidopsis* (*Thompson et al., 2010*).

## Conserved motifs analysis of the MsMATE transporters

The composition of conserved motifs in MsMATE proteins was predicted to illustrate the protein structure (Fig. 3). The motifs presented in MsMATEs ranged from 1 to 11, and the length of the motifs ranged from 15 to 50 aa (Fig. S1). The types of motifs are similar amongst the G2, G3, and G4 groups however differ drastically from the G1 group. The MsMATE proteins in group G1 commonly have five sorts of motifs, and most of them are in the N-terminal, which is less than that of other three groups, such as MsMATE11, 12, 13 and 22, which have only one type of motif. The motif 9 was specifically found in group GI, except for MsMATE22. Motif 9 of G1 group proteins were further determined using the Motif Alignment & Search Tool, the ensemble plant genomes protein of *Arabidopsis*, rice, *M. truncatula* and soybean as database to search against motifs 9, and sort of the sequences by the best combined match to all motifs. Results showed that the motif 9 was identified in four model plants, and all of them belong to MATE transporter family. Based on the phylogenetic tree analysis, Al detoxification/iron

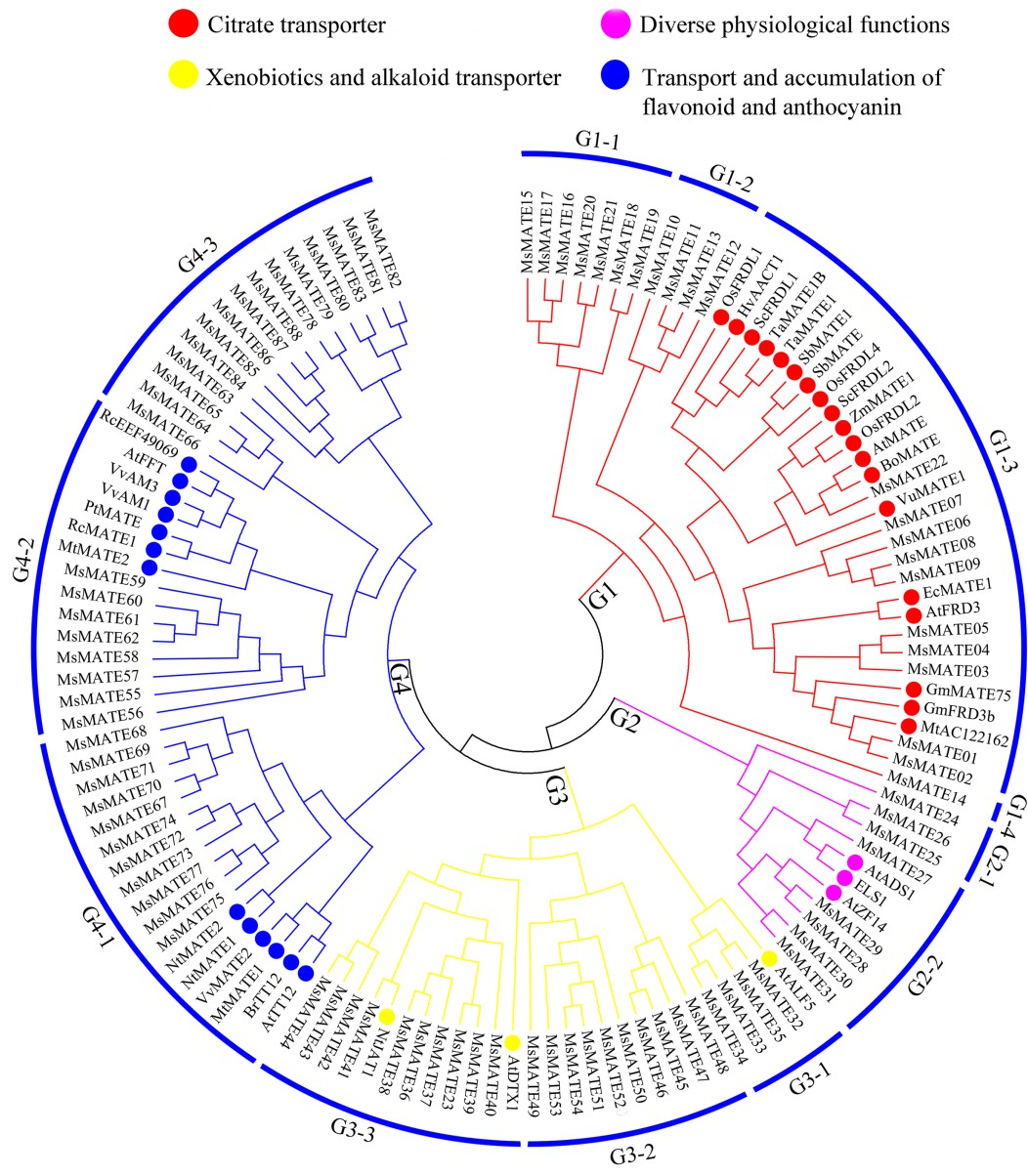

**Figure 2 Functional annotation of MATE proteins from alfalfa and other plants.** The phylogenetic tree was constructed using MEGA7.0 and the neighbour-joining method with 1,000 bootstrap replicates. The tree was divided into four subfamilies comprising 12 smaller subgroups.

translocation-related genes (19 genes) were classed into this groups, implying that motif 9 is essential and specific to MATE transporter family, and might play a crucial role in Al detoxification in alfalfa.

## Functional identification of the MsMATE transporters

Four public databases, Phytozome, GO, KEGG, and UniProt Knowledgebase, were used to predict MsMATE protein annotations with an e-value cut-off of 1e−10. The annotation results are listed in detail in Table S6. The expected MsMATE proteins possibly take

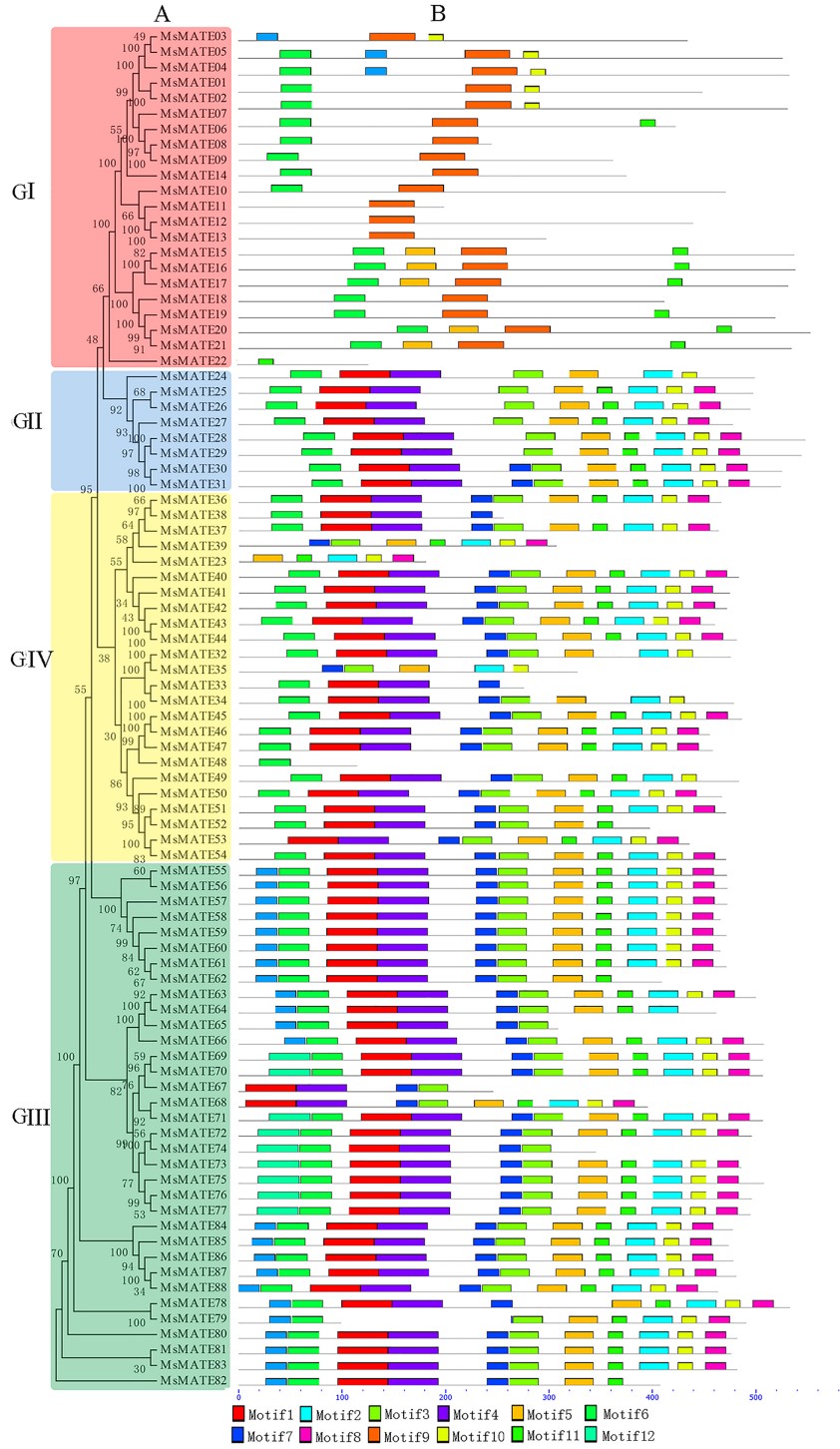

**Figure 3 Phylogenetic relationships and domain compositions of the MsMATE proteins.** (A) The unrooted phylogenetic tree was constructed with 1,000 bootstrap replicates based on a multiple alignment of 88 *MATE* genes amino acid sequences. The four major subgroups are marked with different-coloured backgrounds. (B) The conserved motifs in the MsMATE proteins were identified using MEME. Grey lines represent the non-conserved sequences. Each motif is indicated by a coloured box numbered at the bottom. The length of the motifs in each protein is exhibited proportionally.

apart in xenobiotic efflux, accumulation of secondary metabolites (alkaloids and flavonoids), metal translocation and detoxication, and plant hormone signalling and growth regulation.

In this study, 10 MsMATE proteins predicted to participate in Al detoxification were used to determine the functional and physical relationship through rice and *Arabidopsis* association model using the STRING software, respectively (Fig. S2). Of these 10 MsMATE proteins, the homologous gene suits the easiest bit score by the way of default. OsFRDL4, at the centre of the network node, is an Al-induced citrate transporter that is involved in Al-induced citrate secretion in rice root cells (Fig. S2A) (*Yokosho, Yamaji & Ma, 2011*). The Nramp aluminum transporter (NRAT1) encodes a transporter involved in Al uptake from the root tip cell wall into the cell and plays a key function in rice aluminum tolerance (*Li et al., 2014*; *Xia, Yamaji & Ma, 2011*). The Al-induced gene STAR1 encodes an ATP-binding protein and a TM domain protein, which are required for Al tolerance (*Huang et al., 2009*). The OsFRDL4 could potentially interact with at least seven proteins, including (NRAT1 and STAR1), indicating that it plays important role in the regulation Al-induced citrate transporter (Fig. S2A). Furthermore, the dicot plant model of *Arabidopsis* was further determined (Fig. S2B). Among the 10 MsMATE proteins, MsMATE01, 02, 03, 04, and 05 shared a high similarity to Ferric redictase defective3 (FRD3), which shown to be a citrate efflux transporter, the ectopic expression of FRD3-GFP possessed an enhanced resistance to aluminium in *Arabidopsis* roots (*Durrett, Gassmann & Rogers, 2007*). Meanwhile, MsMATE22 shared the highest similarity (74%) with AT1G51340, previous studies shown that this protein might be involved in the citrate exudation into the rhizosphere to protect roots from Al toxicity (*Liu et al., 2012*, *2009*).

## Expression profiles of MsMATE genes in different tissues

The expression profiles of all 88 *MsMATE* genes at six developmental stages, namely, elongating stem internodes (ES), developing flowers, mature leaf, nitrogen fixing nodules, post-elongating stem internodes (PES) and whole root, were analysed using Illumina RNA-seq data (Fig. 4). Depending on the hierarchical clustering, the *MsMATE* genes have various transcript levels during alfalfa development. Among the 88 *MsMATE* genes, five (*MsMATE 05*, *22*, *31*, *38,* and *82*) showed an overall coverage among all developmental stages (value >1.27). In contrast, the expression level of 13 *MsMATE* genes (*MsMATE 03*, *06*, *14*, *36*, *45*, *51*, *52*, *53*, *60*, *67*, *72*, *80,* and *83*) was relatively low in all tissues (value <1.27) Moreover, among the remaining 70 *MsMATE* genes, 39, 35, 29, 23, 22, and 14 showed high expression levels (value >1.27) in the mature leaf, developing flowers, nitrogen fixing nodules, ES, whole root, and PES, respectively, implying they may play important roles in the corresponding tissues. Although some *MsMATE* genes were classified into one subgroup, the distinct tissue expression profiles of these genes in alfalfa showed that they may have exceptional physiological roles. Furthermore, some *MsMATE* genes had tissue-specific expression, for example, *MsMATE 02, 34*, and *64* in the developing flowers; *MsMATE 01, 25, 26, 46, 62*, and *63* in the whole root; and *MsMATE 15, 42, 43, 55, 56, 59, 76, 77*, and *78* in nitrogen-fixing nodules.

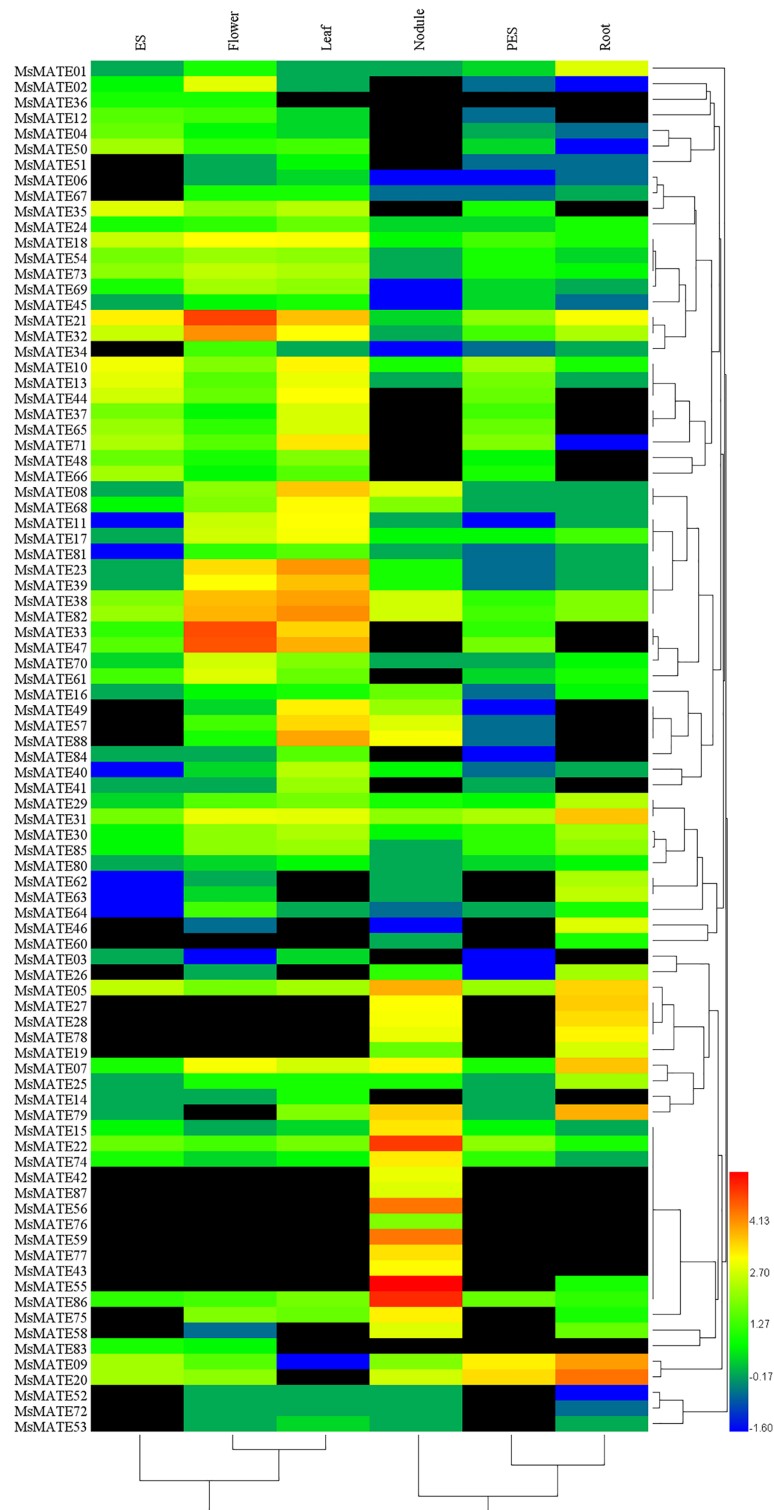

**Figure 4 Heatmap representation and hierarchical clustering of the *MsMATE* genes in various tissues of alfalfa.** The transcript data of six tissues were used to re-construct the expression patterns of *MsMATE* genes. The black boxes indicate that the transcript abundance is zero. The bar at the bottom of the heat map represents the relative expression values; values <1.27 represent down-regulated expression and values >1.27 represent up-regulated expression.

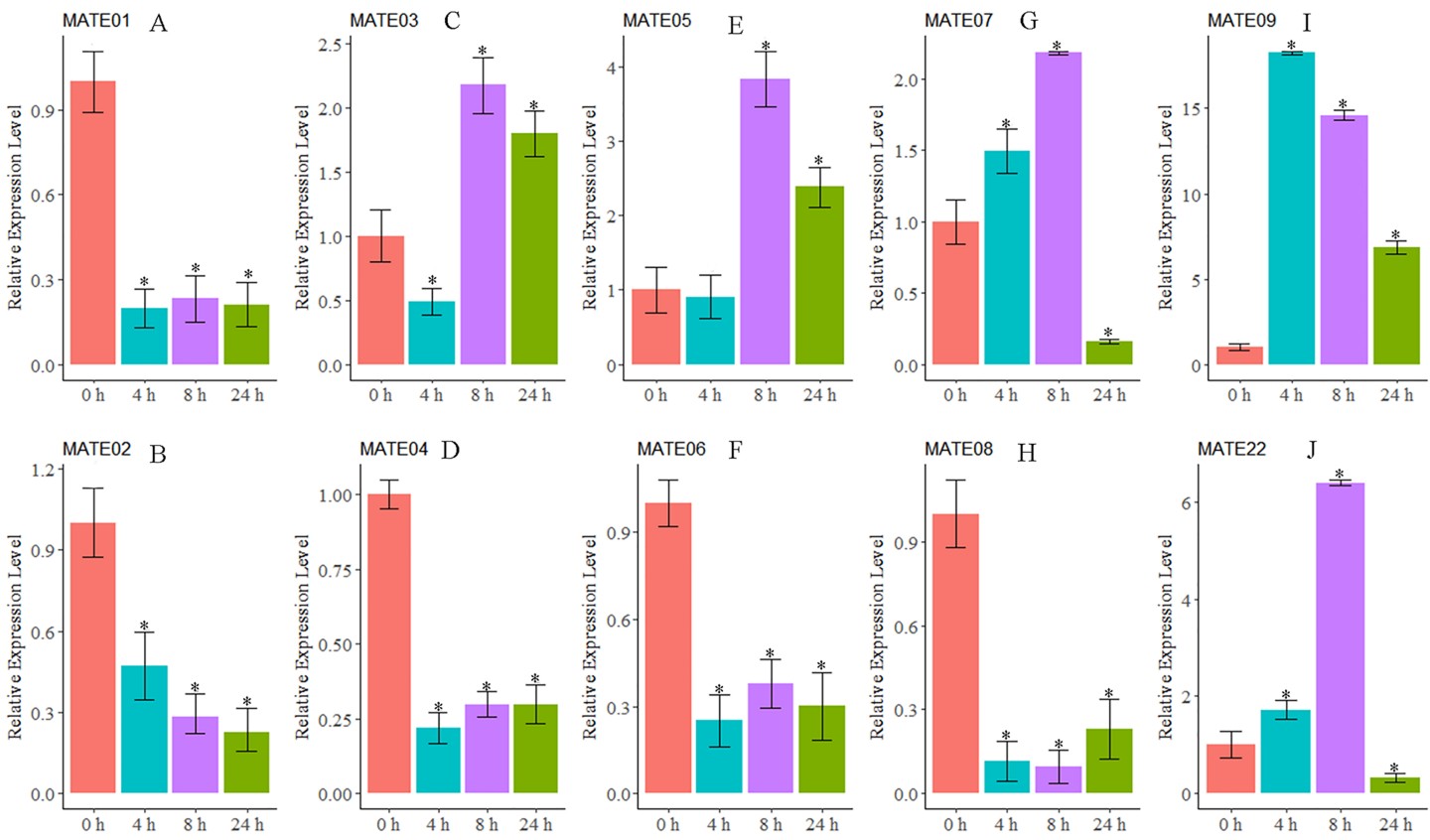

**Figure 5  The relative expression ratio of ten representative MsMATE genes in Al stress have been calculated with reference gene.** (A–H) indicate significant differences between different treatment times (*$p < 0.05$). The name of the gene is written on the top of each bar diagram (error bars indicate the standard deviation from three replicates). (A) MATE01, (B) MATE02, (C) MATE03, (D) MATE04, (E) MATE05, (F) MATE06, (G) MATE07, (H) MATE08, (I) MATE09 and (K) MATE22. 0 h, 4 h, 8 h and 24 h represented Al treatments for 0, 4, 8 and 24 hours, respectively.

### Expression of the subgroup G1–3 MsMATE genes in response to Al toxicity

As we were interested in the roles of *MATE* genes involved in Al stress in alfalfa, the expression patterns of 10 MsMATEs that were predicted to be Al detoxification-related based on the sequence similarity were further analysed using qRT-PCR. As shown in Fig. 5, *MsMATE05* and *MsMATE09* were significantly up-regulated after Al treatment. The expression of *MsMATE03, MsMATE07,* and *MsMATE22* fluctuated across early (4 h), middle (8 h) and/or late (24 h) Al exposure time points. Among these five up-regulated *MsMATE* genes, *MsMATE09* has the highest expression level, nearly up-regulated 18-fold at time point 4 h compared with control. In contrast, the expression of the five remaining genes (*MsMATE01, MsMATE02, MsMATE04, MsMATE06,* and *MsMATE08*) was significantly down-regulated after Al treatment.

### DISCUSSION

Aluminum toxicity is a considerable hindrance of crop comprehensive completion in acidic soils worldwide. At pH values beneath 5, Al tends to be dissolved as $Al^{3+}$ ions, which

are quite toxic to plant roots and can further limit crop production (*Kochian et al., 2015*; *Vonuexkull & Mutert, 1995*). Thus, a sound comprehension of the mechanistic basis and gene quality giving protection of Al poisonous quality has been seriously explored over the final decade. To date, two major elementary sorts of Al resistance mechanisms are characterised: Al exclusion mechanisms and Al tolerance mechanisms. Among them, the Al-induced unharness of organic compounds preventing the contact of phytotoxic $Al^{3+}$ with the Al-sensitive root apex is the best-documented mechanism (*Kochian et al., 2015*). The MATE family is very important cluster of multidrug effluence transporters that square measure liable of Al resistance support root citrate exudation in response to Al stress. Some necessary legume crops, like soybean and alfalfa, are sensitive to acidic soils as well as $Al^{3+}$ environment; therefore, their development and yield could be severely restricted (*Wang et al., 2017*; *Liu et al., 2016*). However, restricted data is presented for alfalfa response to $Al^{3+}$ stress, for the most part of their complicated biological science, massive genomes and complicated resistance mechanisms. Currently, a total of 112,626 unigenes is publicly available for download and exploration on the AGED database (*O'Rourke et al., 2015*), which provides an opportunity for MsMATE transporter screening and characterisation in alfalfa.

In our study, a complete of 88 MsMATE transporters were finally known in alfalfa and this number is 1.5- and 1.6-fold that of *Arabidopsis* and rice respectively, that is in line with the magnitude relative of the full variety of transporter sequences in alfalfa thereto in genus *Arabidopsis* and rice (*Benedito et al., 2010*). However, as the genome sequence of alfalfa has not been released, the members of the *MsMATE* gene identified in this study may not be fully covered. To review the organic process relationship between the MsMATE transporters and the MATE transporters from completely different plant species, all MsMATEs and MATE transporters from the monocot (rice) and dicot (*Arabidopsis*) model systems were subjected to phylogenetic analyses. As illustrated in Fig. 1, the organic process tree categorised the MsMATE transporters into four completely different groups along with their AtMATE and OsMATE orthologs, and 10 MsMATE transporters were clustered into subgroups G1–3 with 19 previously reported Al detoxification genes (Fig. 2). The phylogenetic scrutiny is one of the fastest, simplest, and relatively most precise way to predict gene function that may want to be subsequently prioritised for further practical confirmation. Previous studies have provided strong evidence for function prediction of gene families based on phylogenetic analysis, including MATE transporters (*Chen et al., 2015*; *Le et al., 2011*; *Liu et al., 2016*). These 10 *MsMATE* genes might be potential candidate genes involved in Al tolerance in alfalfa and warrant further analysis.

Gene structure analysis shows that different MsMATE members in alfalfa shared a relatively conserved motif distribution tend to cluster together. As shown in Fig. 3, MsMATEs in groups GII, GIII, and GIV had most types of MATE motifs, indicating that groups GII, GIII, and GIV might have diverse functions. MsMATEs in group GI only have five kinds of motifs, and most of them are in the N-terminal, which is less than that of the other groups. It is interesting to note that motif 9 was only found in group GI, except for MsMATE22. *Ligaba et al. (2013)* and *Sasaki et al. (2014)* characterised the useful, structural and organic process nature of domains underlying Al sensitivity of Al-activated

malate/anion transporters (TaALMT1), and indicated that the N-domain relies to make conductive pathway and mediate particle transport even within the absence of the C-domain. The GI group transporters were predicated to be Al detoxification-related based on the phylogenetic tree analysis, implying that motif 9 might play a crucial role in Al detoxification in alfalfa. Previous research has offered a good vary of references that amino acid motifs and functionally important domains are concerned in diverse biological processes, like transcriptional activity, protein–protein interactions, and nuclear localisation (*Liu, White & Macrae, 1999*). These motifs or domains are typically preserved among members of a subgroup of giant families, moreover proteins of those motif are additional possible to share similar functions. To date, the exact roles of these conserved motifs in MsMATEs have not been reported. The next step of functional identification of these motifs would benefit our considerate of the structure-function relationship of MsMATEs.

The identification of the expression profiles of genes, especially those with tissue-specific, will be useful for classification of genes that are concerned with the regulation of the precise nature of individual tissue. Increasing lines of evidence suggested that overexpression of tissue-specifically genes can promote tissue remodelling and functional improvement (*Shitan et al., 2014*; *Shoji et al., 2009*). Given the different tissue-specific expression and varying subcellular localisations (Table S4), the MATEs identified here might possess diverse functions during various physiological and cellular processes in alfalfa. For example, *AtDTX50*, a MATE homologous gene of *Arabidopsis*, was expressed primarily in vascular tissues and guard cells and was powerfully up-regulated by exogenous abscisic acid in leaves (*Zhang et al., 2014*). *M. truncatula* MATE02, which is uttered mostly in leaves and flowers, shows advanced transport ability for anthocyanin pigmentation and pale flower colour, however, lower potency for alternatives glycosides (*Zhao et al., 2011*). Recently, it had been discovered in genome-wide expression analysis that *OsMATE1* and *OsMATE2* genes altered growth and morphology in transgenic *Arabidopsis*, suggesting their potential roles in regulating plant expansion in transgenic lines (*Tiwari et al., 2014*). In this study, the gene translation of the MsMATEs showed a high inconsistency in transcription plenty (Fig. 4). Some genes exhibited constituted expression profiles, such as *MsMATE 05*, *22*, *31*, *38*, and *82*, which were expressed in all examined tissues, while some others were found to be expressed specifically high only in the root (*MsMATE 01*, *25*, *26*, *46*, *62*, and *63*), flower (*MsMATE 02*, *34*, and *64*) and nodule (*MsMATE 15*, *42*, *43*, *55*, *56*, *59*, *76*, *77*, and *78*). The various expression profiles of the *MsMATE* genes implicated their functional diversity, which could provide useful guidance for functional identification of candidate genes for specific traits in genetic engineering of alfalfa.

Al quickly affects a variety of cellular processes and quickly inhibits cell elongation in the root, which is taken into account to be the first target of Al stress (*Chandran et al., 2008a*, *2008b*; *Ryan, Ditomaso & Kochian, 1993*). In our study, 10 *MsMATE* genes that were predicated to be Al detoxification-related based on the phylogenetic analysis were chosen for qRT-PCR confirmation. The results proved that the expression of five genes (*MsMATE03, MsMATE05, MsMATE07, MsMATE09,* and *MsMATE22*) in the

root tip after Al stress was significantly induced (Fig. 5), signifying that these genes might play a vital role in conferring Al tolerance in alfalfa and so would be attention-grabbing candidate genes for more determination. The *MsMATE03*, *MsMATE07*, *MsMATE09*, and *MsMATE22* shared 64%, 93%, 97%, 89% identity with a putative *M. truncatula* *MtMATE66* (*Medtr2g097900.1*) respectively. The root growth of the *mtmate66* mutant was less than that of wild type under $Al^{3+}$ treatment, and chlorotic was observed under Fe-deficient condition in seedlings. Overexpression of *MtMATE66* rendered hairy roots more tolerant to $Al^{3+}$ toxicity (*Wang et al., 2017*). *MsMATE03* and *MsMATE05* shared 65% and 67% identity with a putative *VuMATE1* (Vigna umbellate MATE1), respectively (*Fan et al., 2014*). The expression of these two genes up to the highest-level by 8 h of exposure to $Al^{3+}$ stress. Previously study showed that the expression of *VuMATE1* was increased along with the increaseed external Al concentrations after 4 h of exposure, indicated that these two genes may function as early Al-responsive transporters in root. The ortholog of *MsMATE03* and *MsMATE05* in *Arabidopsis* is FRD3, which shown to be a citrate efflux transporter. The plants overexpressed with *ATFRD3* had significantly higher amounts of citrate in their root exudates and possessed an enhanced resistance to aluminium compared to untransformed controls (*Durrett, Gassmann & Rogers, 2007*). Previously research shown that the sequence similarity-based methodology had an associate accuracy rate of 83% for stress-related GmNAC sequence identification of soybean (*Le et al., 2011*). So, these results in our study would provide useful information by identifying candidate Al stress related *MsMATE* genes and to engineer alfalfa plants for enhanced Al stress resistance.

Furthermore, MATE family members may possess diverse functions in plant cellular and physiological processes because of their different subcellular localisations, substrates, and tissue-specific of gene translation. Interestingly, some genes in the MATE family, such as *EDS5* and *ADS1* in *Arabidopsis*, can proceed as a positive or negative regulator in response to disease resistance (*Ishihara et al., 2010*; *Sun et al., 2011*). In our study, the expression patterns of five *MsMATE* genes (*MsMATE01*, *MsMATE02*, *MsMATE04*, *MsMATE06*, and *MsMATE08*) were also found to be repressed after Al treatment in alfalfa root tips (Fig. 5). However, the specific reasons for this down-regulation of these genes are still unclear, and whether these *MsMATE* genes can function as negative regulators in Al stress in alfalfa still needs to be further investigated.

## CONCLUSIONS

To investigate this study, we tend to perform a comprehensive transcriptome-wide survey of MsMATE transporters in an important legume species, alfalfa. A complete of 88 transporters were categorised and classified into four main classes comprising 11 smaller subgroups consistent with organic process interaction and structural characteristics. The expression profile of *MsMATE* genes in various tissues/organs indicates that this gene family is wide concerned in alfalfa upgrading. The expression of *MsMATE03, MsMATE05, MsMATE07, MsMATE09,* and *MsMATE22* was significantly induced after Al treatment, indicating that these genes would possibly play a vital role in conferring

Al tolerance in alfalfa. The elaborate results conferred here would offer valuable data for the useful investigation and application of this transporter family in alfalfa.

### Funding

This research was supported by the Program for Changjiang Scholars and Innovative Research Team in University (IRT_17R50), the National Natural Science Foundation of China (31502000), the 111 project (B12002) and the Fundamental Research Funds for the Central Universities (lzujbky-2016-8). The funders had no role in study design, data collection and analysis, decision to publish, or preparation of the manuscript.

### Grant Disclosures

The following grant information was disclosed by the authors:
Program for Changjiang Scholars and Innovative Research Team in University: IRT_17R50.
National Natural Science Foundation of China: 31502000.
111 project: B12002.
Fundamental Research Funds for the Central Universities: lzujbky-2016-8.

### Competing Interests

The authors declare that they have no competing interests.

### Author Contributions

- Xueyang Min conceived and designed the experiments, performed the experiments, analysed the data, contributed reagents/materials/analysis tools, prepared figures and/or tables, authored or reviewed drafts of the paper, approved the final draft.
- Xiaoyu Jin performed the experiments, analysed the data.
- Wenxian Liu conceived and designed the experiments, analysed the data, contributed reagents/materials/analysis tools, prepared figures and/or tables, authored or reviewed drafts of the paper, approved the final draft.
- Xingyi Wei performed the experiments.
- Zhengshe Zhang performed the experiments.
- Boniface Ndayambaza prepared figures and/or tables, authored or reviewed drafts of the paper.
- Yanrong Wang contributed reagents/materials/analysis tools, authored or reviewed drafts of the paper, approved the final draft.

### Data Availability

  The raw measurements are available in Supplementary Files.

### Supplemental Information

Supplemental information for this article can be found online at http://dx.doi.org/10.7717/peerj.6302#supplemental-information.

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
