# Peer review of "Transcriptome-wide characterization and functional analysis of MATE transporters in response to aluminum toxicity in Medicago sativa L"

_PeerJ, doi:10.7717/peerj.6302_

## Round 0.1 · original submission · Major Revisions

Please address the numerous queries by our reviewers. Additionally, your manuscript requires an extremely thorough language and grammar check by a professionally-proficient English speaker.

Reviewer 1 ·

Basic reporting

No comment

Experimental design

No comment

Validity of the findings

No comment

Additional comments

The study has identified 88 non-redundant MATEs in Alfalfa (Medicago sativa L.) based on BLAST analysis and characterized their gene structures, phylogenetic relationship, and expression patterns in different tissues. Ten MsMATEs considered as Al-stress related were furthered validated their expression profiles under Al stress treatment. The manuscript is generally well-written and easy to read for the materials and methods and results sections. But there are gramma or spelling mistakes in the introduction and discussion sections, and many sentences in the discussion section are hard to read and understand. The representation of the materials and methods is clear, but some information is missing, such as the transcriptome data used for expression patterns of MsMATEs. The result section has included the text for data interpretation, while the discussion section contained some wrong description of the results and has many sentences not easy to read. For these reasons, the present MS certainly warrants thorough revision before consideration of publication in Peer J.

Major comments:
1) In the introduction section, the phrase “Al ability” has appeared many times, but it is misleading. Does it mean antioxidant ability against aluminum toxicity? In L54, should the phrase “no floxacin” be corrected as ”nofloxacin”? In L57 to 59, five species have been listed, but only four MATE transporters gene numbers have been shown. L94 to 96, the last sentence of this paragraph is hard to read.
2) In the Materials and methods section, there are missing data needed. In L99, the alfalfa cultivar name in incomplete. In L101, full name for MS should be provided in its first appearance in the text. In L103, is there missing information for temperature in the brackets? In L108, do you mean that e-value cutoff is 1e-5? According to the text in the result section (subtitle: Expression profiles of MsMATE genes in different tissues), the datasets used for RNA-seq analysis was not mentioned in the Materials and Methods section.
3) In the Result section, the text for data interpretation is included in this section, which is not suitable because a separate discussion section is existed in the manuscript. For Figure 1 and Figure 2, no bootstrap values have been displayed. Under the subtitle “Functional identification of the MsMATE transporters”, the text is not informative and litter information has been provided in Figure 4. It seems to me that this part of result is no different from gene annotation and the gene network for rice is not suitable to display here. Maybe, an improved version of this figure could be listed as a supplementary figure. In L260 to 261, the sentence “Increasing of the evidence….” should be rewrote and the whole paragraph should be moved to the discussion section. In L269 to 270, the gene names “MsMATE05” and “MsMATE09” were not in italics.
4) In the discussion section, many sentences are hard to read. In L278, the sentence “…, that can be act as phytotoxic…” should be rewrote; In L353, the mistake in the sentence “it’s been urged…”should be corrected. In L294 to 295, the sentence is hard to read. In L295, “the genome of alfalfa….” should be corrected as “the genome sequence of alfalfa….”. In L355 to 356, It is hard to conclude that five MsMATE genes is Al stress related. As we seen from RT-qPCR results, all genes were responsive to Al stress treatments. It will be great if homologous gene related with Al stress in model plants or reported researches could be discussed here and comparing with the results of this study. The whole discussion section should be carefully revised.

Reviewer 2 ·

Basic reporting

Language of the manuscript should be improved and also use proper scientific words.
Excellent background information has been provided by citing correct references.
Article can be improved on result part especially when talks about conserved motifs where readers fail to get clarity.
What was the criteria for selecting a subset of MATE genes for Al responsive expression analysis? A more clear explanation can be given here.

Experimental design

Methods have been described in detail.

Validity of the findings

Data is good but a more concrete conclusion and future prospects need to be done.

Additional comments

A Well executed experiment with proper documentation. But try to put your findings in a more interesting scientific language by reducing text and concentrating in giving accurate well defined point based explanation

Reviewer 3 ·

Basic reporting

Could you confirm citation in your paper

Line 53-57; A MATE gene (Norm) was cloned in Vibrio parahaemolyticus, which possesses an energy-dependent system that effluxes no floxacin and other antimicrobial agents outside of the cells by Na+/drug antiport (Morita et al. 1998; Yokosho et al. 2016)
This reference is correct ? I think Yokosho et al.,2016 described rice MATE transporter.

Line 180-181; To similarly predict and distinguish the feature of identified MsMATE genes in this study, an NJ tree was constructed using 38 well-known plant MATE transporters (Eisen 1998; Takanashi et al. 2015; Tiwari et al. 2014) and 88 MsMATE proteins (Fig. 2, Table S5).
And I think “Eisen 1998” this citation is not necessary.

Line 235-238 The Nramp aluminum transporter (NRAT1) encodes a transporter involved in Al uptake from the root tip cell wall into the cell and plays a key function in rice aluminum tolerance (Li et al. 2014). I think Xia et al., 2011 also add citation.
The Al-induced gene STAR1 encodes an ATP-binding protein and a transmembrane domain protein, which are required for Al tolerance (Arenhart et al. 2014).
Arenhart et al., 2014 is not described function of STAR1. Author should be cite Huang et al., 2009

Experimental design

I have some question in experimental design and suggestion for improve expression analysis part.


Line 230-231; In this study, ten MsMATE proteins predicted to participate in Al detoxification were used to determine the functional and physical relationship through a rice association model using the STRING software (Fig. 4). Why author used rice association model. Since Alfalfa is dicot plants, I think it is better to Arabidopsis data.

Line 180-181; To similarly predict and distinguish the feature of identified MsMATE genes in this study, an NJ tree was constructed using 38 well-known plant MATE transporters (Eisen 1998; Takanashi et al. 2015; Tiwari et al. 2014) and 88 MsMATE proteins (Fig. 2, Table S5). Why author select 38 genes and what do you mean “well-known plant MATE” At least Takanashi et al., 2015 (2014?) mentioned 44 genes.


I think author should be more detail expression analysis for characterization of these MATE genes. 1) To confirm Al specific expression, author should be check to response in different pH, other metal stress. Because Arabidopsis AtMATE1 response to Al and low pH. 2) To determine highest expressed gene in root tip, author compare absolute expression level in 10 candidate genes. 3) Comparison of different root segment. In this paper Author determined expression at only root tip. I think Al tolerance citrate transporter expressed in root tip. On the other hand citrate transporter for iron translocation are expressed in basal root region. So this is more information for understanding of 10 MATE gene function.

Validity of the findings

Figure 2; Author described G1; Al detoxification/iron translocation. It should be citrate transporter. G3; Efflux of various compounds. It is also incorrect caption. Because almost all MATE is efflux transporter. And G3 group including NtJAT1. This protein tansport nicotine, which is one of the alkaloid. Could you consider caption of G1,G2, G3, G4 in Fig 2.

Line 57-59; To date, 56, 53, 117 and 70 MATE transporters have been identified from Arabidopsis thaliana (Li et al. 2002), Oryza sativa (Tiwari et al. 2014), Glycine max (Liu et al. 2016), Medicago truncatula (Wang et al. 2017), and Zea mays (Zhu et al. 2016), respectively. This number is different in Line 113-114. And the number of Zea mays is including ?

Line 164-166; The ORFs of these genes 165 varied in length from 321 (MsMATE80) to 1788 (MsMATE10) bp. This length is different from your data. Supplemental data showed that MsMATE80 is 1485bp, MsMATE10 is 1449 (Sup data1).

Additional comments

This paper re-construct genome-wide analysis for identify MATE transporter in Medicago sativa L , which is an important crops, but sensitive to acidic soil condition. Author identified 88 MATE genes, which were divided 4 subfamilies comprising 11 subgroups. As a result of expression analysis by RT-PCR, they found 10 candidate MATE genes for Al tolerance mechanism in Alfalfa.
But I think more detail expression analysis is improve your speculation (described above).
And this author has some misunderstanding for reference paper. Author should be more carefully cite reference paper.

Other minor point

Line 262 (Nobukazu Shitan 2014; Tovkach et al., 2013) should be Shitan et al., 2014
Line 103 (, light intensity 120 μmol) remove “,”.
Table S4 MsMATE33 (Cell H35) should be “plas”

---

## Round 0.2 · accepted · Accept

I am happy to approve your manuscript for publication in PeerJ.

# Reviewer 1 ·

Basic reporting

No comment.

Experimental design

No comment.

Validity of the findings

No comment.

Additional comments

The authors have performed the suggested revsions in the materialas and methods section, results section and discussion section. Since the authors have addressed all major concern, the present MS is acceptable in my opinion.

Reviewer 3 ·

Basic reporting

no comment

Experimental design

no comment

Validity of the findings

no comment

Additional comments

This paper re-construct genome-wide analysis for identify MATE transporter in Medicago sativa L , which is an important crops, but sensitive to acidic soil condition. Author identified 88 MATE genes, which were divided 4 subfamilies comprising 11 subgroups. As a result of expression analysis by RT-PCR, they found 10 candidate MATE genes for Al tolerance mechanism in Alfalfa. I think this is beneficial data for understanding of molecular Al tolerance mechanism in Medicago sativa in future.